# Calcium’s Role in Orchestrating Cancer Apoptosis: Mitochondrial-Centric Perspective

**DOI:** 10.3390/ijms24108982

**Published:** 2023-05-19

**Authors:** Dong-Oh Moon

**Affiliations:** Department of Biology Education, Daegu University, 201, Daegudae-ro, Gyeongsan-si 38453, Gyeongsangbuk-do, Republic of Korea; domoon@daegu.ac.kr; Tel.: +82-53-850-6992

**Keywords:** calcium, apoptosis, ER stress, mitochondria

## Abstract

Calcium is an essential intracellular messenger that plays a vital role in controlling a broad range of cellular processes, including apoptosis. This review offers an in-depth analysis of calcium’s multifaceted role in apoptosis regulation, focusing on the associated signaling pathways and molecular mechanisms. We will explore calcium’s impact on apoptosis through its effects on different cellular compartments, such as the mitochondria and endoplasmic reticulum (ER), and discuss the connection between calcium homeostasis and ER stress. Additionally, we will highlight the interplay between calcium and various proteins, including calpains, calmodulin, and Bcl-2 family members, and the role of calcium in regulating caspase activation and pro-apoptotic factor release. By investigating the complex relationship between calcium and apoptosis, this review aims to deepen our comprehension of the fundamental processes, and pinpointing possible treatment options for illnesses associated with imbalanced cell death is crucial.

## 1. Introduction

Calcium, an essential second messenger in cells, controls numerous cellular functions, including cell death [1]. The concentration of calcium varies in different cell compartments, with the cytoplasm maintaining relatively low levels to ensure effective cellular signaling and prevent cytotoxic effects. This balance is achieved through various mechanisms, such as calcium pumps, sodium–calcium exchangers, calcium-binding proteins, and the mitochondrial calcium uniporter (MCU), that help regulate cytoplasmic calcium levels [2]. Cancer cells are characterized by modifications in calcium channel, pump, and binding protein function, leading to calcium concentrations that surpass the usual limits seen in healthy cells. This calcium surplus fosters cell growth and malignancy. Elevated intracellular calcium levels in many cancer types can be attributed to heightened activity or irregular control of specific calcium channels and pumps. Examples include the overactivity of certain members of the transient receptor potential (TRP) channel family, like TRPM3, TRPC1, TRPC6, TRPV4, and TRPV6, seen across a variety of tumors [3]. In addition, channels like TRPA1 are notably more active in cancers such as those of the breast and lung [4]. Regulated Cell Death (RCD), which includes apoptosis, autophagy, necrosis, etc., is vital for maintaining homeostasis in living organisms, and calcium signaling plays a pivotal role in controlling it [5]. Calcium’s central role in apoptosis is particularly significant due to its influence on the key molecular events and signaling pathways determining cell fate. Understanding calcium’s function in cell death can potentially uncover new therapeutic targets for illnesses related to dysregulated cell death.

## 2. The Role of Calcium in Mitochondria-Derived Apoptosis

Apoptosis, a form of regulated cell death, is essential for tissue development and maintenance. It can be activated through two primary routes—intrinsic and extrinsic pathways—both of which can interact and converge at the level of effector caspases [6,7]. The extrinsic pathway begins when death-inducing molecules like Fas ligand (FasL), tumor necrosis factor (TNF), or TNF-related apoptosis-inducing ligand (TRAIL) bind to corresponding cell surface death receptors. This binding initiates the assembly of adaptor proteins, which activate the initiator caspase, caspase-8. The activated caspase-8 can directly activate effector caspases or cleave the pro-apoptotic B-cell lymphoma 2 (Bcl-2) family protein Bid, thus connecting the extrinsic pathway to the intrinsic pathway. The intrinsic pathway is triggered by internal cellular stress signals, such as DNA damage or oxidative stress. These signals can disrupt the balance between pro-apoptotic (e.g., Bax, Bak, Bad) and anti-apoptotic (e.g., Bcl-2, Bcl-xL) Bcl-2 family proteins. The activation of pro-apoptotic proteins results in mitochondrial outer membrane permeabilization (MOMP), allowing the release of cytochrome c and other pro-apoptotic factors from the mitochondria. Cytochrome c then interacts with apoptotic protease activating factor-1 (Apaf-1) and pro-caspase-9 to form a complex known as the apoptosome, which activates caspase-9. Finally, caspase-9 triggers the effector caspases, leading to apoptosis.

### 2.1. Mitochondrial Calcium Uptake from ER

Calcium signaling begins with the release of calcium ions from internal reservoirs, like the ER. Mitochondria-associated membranes (MAMs) are specialized microdomains where the ER and mitochondria come into close contact [8,9]. MAMs play essential functions in diverse cellular activities, such as fat metabolism and calcium equilibrium, and communication between the two organelles. The space separating the ER and the mitochondrial outer membrane can vary between 10 and 100 nm [10,11]. Typically, the gap is narrower at the smooth endoplasmic reticulum (10–15 nm) compared to the rough end (20–30 nm). These variations in distance may be associated with the presence of ribosomes [12,13]. Despite their nearness, the pair of membranes do not merge, and the contact is maintained through protein–protein interactions involving tethering complexes. Several protein complexes and tethering factors have been discovered that contribute to the generation and stabilization of MAMs.

Initiation of calcium movement from the ER to the mitochondria occurs via Inositol 1,4,5-trisphosphate receptors (IP_3_Rs)-Grp75-VDACs-MCU Complex. IP_3_Rs are integral membrane proteins that play a critical role in calcium signaling within cells. In mammals, the expression of three distinct isoforms of IP_3_R (IP_3_R1, IP_3_R2, and IP_3_R3) is observed, each exhibiting diverse IP_3_ affinities and found in different amounts across various cell types. They are primarily found on the ER membrane and function as ligand-gated calcium channels. When IP_3_ binds to IP_3_Rs, they facilitate the release of calcium ions from the ER into mitochondria in MAMs, which can trigger various cellular processes [14,15]. 

Anion channels known as Voltage-dependent anion channels (VDACs) can be found on the outer mitochondrial membrane and have an impact on a variety of cellular functions, including apoptosis, metabolism, and calcium regulation [16,17]. In mammals, three VDAC isoforms exist, with VDAC1 being the predominant form in most cell types. Its structure, confirmed by NMR and X-ray crystallography, consists of 19-stranded β-barrels that form a pore roughly 1.5 × 1 nm in size [17]. The VDAC open state has a 3 nm pore diameter, allowing the passage of calcium, ATP, and other metabolites [18]. Transient overexpression of VDAC in various cell types has been shown to increase mitochondrial calcium concentrations [19]. In addition, also overexpression of VDACs has been shown to strengthen the connection between the ER and mitochondria, enhancing calcium flux from the ER to the mitochondria [20]. Conversely, silencing VDAC1 reduces the interaction between Glucose-regulated protein 75 (Grp75) and IP_3_R1, indicating a decrease in ER-mitochondria communication [21].

Grp75 connects IP_3_Rs and VDACs, helping maintain the structure of MAMs [22]. When there is an overproduction of Grp75, the frequency of interaction sites between IP_3_R1 and VDAC1 increases [23]. On the other hand, reducing the expression of Grp75 can also decrease the connection between IP_3_R1 and VDAC1.

Sigma receptors are made up by sigma-1 (Sig-1R) and sigma-2 (Sig-2R) subtypes. Sig-1Rs are enriched at the ER of MAMs and interact with the calcium-binding chaperone BiP [24]. The interaction is contingent upon the level of calcium in the ER, with a decrease in ER luminal calcium causing dissociation and activation of both proteins [24]. During ER stress, Sig-1R can separate from binding BiP and act as a chaperone for IP_3_R in MAMs. In this location, Sig-1R helps to maintain the stability of IP_3_R, thus avoiding its degradation, and enabling continuous release of calcium towards the mitochondria results in an enhancement in ATP production and supports cell survival [25]. However, prolonged Sig-1R activation may result in excessive transfer of calcium to the mitochondria, which has recently been shown to lead to cell death in the brain endothelial cells [26]. Indeed, Sig-1R allows ankyrin to be detached from IP_3_R, which stabilizes and enhances the opening of IP_3_R. 

The Bcl-2 protein family encompasses both pro- and anti-apoptotic proteins, which control the intrinsic pathway of apoptosis. Bcl-2, an anti-apoptotic protein, functions in two manners: by binding and inhibiting pro-apoptotic Bcl-2 family proteins such as Bax and Bak at the mitochondrial level, and by binding and modulating IP_3_R at the ER level [27,28]. Bcl-2 directly interacts with IP_3_R, fostering pro-survival calcium fluctuations and preventing pro-apoptotic calcium release [29,30]. Bcl-2 precisely regulates the uptake of calcium by the mitochondria, avoiding excessive calcium levels in the mitochondria. Bcl-2 can also attract additional proteins that can have an indirect effect on IP_3_R [31]. In many cases of cancer, an increased expression of Bcl-2 helps the cancer cells to escape cell death, making Bcl-2 targeting an attractive anticancer strategy [32]. Bcl-2 ligands, such as BIRD-2 (a BH4 mimetic), have been developed to modulate IP_3_R through Bcl-2, triggering cell death in various cancer models [33,34]. BH3-mimetics are another category of peptide that focus on Bcl-2. These peptides bind to the hydrophobic cleft of Bcl-2, hindering it from connecting with and suppressing the pro-apoptotic proteins Bax and Bak. They do not affect IP_3_R activity [35]. Targeting Bcl-2 holds promise as a potential therapeutic approach for cancer treatment, with one peptide having been approved for treatment.

Ankyrin B (AnkB) is a cytoskeletal protein involved in regulating IP_3_R activity. It has been demonstrated that AnkB interacts with the sigma-1 receptor and IP_3_R, influencing the regulation of IP_3_R activity and the flow of calcium between the ER and mitochondria [36].

Presenilins are a group of proteins that consist of two main variants, presenilin-1 (PSEN1) and presenilin-2 (PSEN2). These proteins play a role in Alzheimer’s disease as part of the gamma–secretase complex [37]. They have been demonstrated to be vital in the regulation of IP_3_R activity and intracellular calcium signaling. PSENs directly interact with all three IP_3_R isoforms (IP_3_R1, IP_3_R2, and IP_3_R3) at the ER, modulating calcium release through modulation of these receptors’ activity [38]. The interactions occur mainly through the N-terminal and loop regions of PSENs and the regulatory and coupling domain of IP_3_Rs. This binding affects the binding affinity of IP_3_Rs to IP_3_, thus inhibiting the calcium release from the ER. Especially the regulation of IP_3_R activity by PSENs is essential for maintaining calcium homeostasis in neurons. When PSENs function is disrupted, as in the case of some familial Alzheimer’s disease mutations, IP_3_R-mediated calcium release can be altered, leading to impaired calcium signaling. This dysregulation may contribute to synaptic dysfunction, neuronal loss, and ultimately the development of Alzheimer’s disease. Additionally, presenilins have been suggested to affect IP_3_R function indirectly by influencing the lipid environment in the ER membrane. Since the activity of IP_3_Rs can be modulated by the lipid composition of the ER membrane, PSENs may play a role in maintaining the optimal lipid environment for IP_3_R function.

Akt, also known as protein kinase B, is a serine/threonine kinase that has a vital role in several cellular processes, including cell survival, growth, and metabolism. Akt has been shown to have an impact on IP_3_R at the ER level. Akt-mediated phosphorylation of IP_3_R has been reported to modulate their activity and impact cellular calcium homeostasis [39]. Specifically, Akt can phosphorylate the IP_3_R at distinct serine residues, leading to a change in the receptor’s sensitivity to IP_3_, thereby influencing the calcium release from the ER. Phosphorylation of IP_3_R by Akt can result in either increased receptor activity and calcium release or suppressed receptor activity, reducing calcium release and preventing excessive cytosolic calcium levels.

The MCU complex mediates calcium transport into the mitochondrial matrix. This process is of utmost importance in controlling mitochondrial function, cellular energy metabolism, and apoptosis. The MCU complex is a highly conserved machinery across various species and is embedded in the inner mitochondrial membrane. The MCU complex comprises several components, including pore-forming subunits MCU and MCUb, regulatory proteins MICU1, MICU2, and MICU3, and accessory proteins EMRE and MCUR1 [40,41]. MCU is the central subunit that forms pores in the MCU complex. It forms a calcium channel with high selectivity, allowing calcium ions to pass through the inner mitochondrial membrane. The MCU has two transmembrane domains, with its N- and C-termini positioned towards the intermembrane space within the mitochondria. The pore that conducts calcium is created by a tetramer of MCU subunits [42,43]. MCUb is an MCU paralog that form hetero-oligomers with MCU, modulating the channel’s activity. MCUb has lower calcium conductivity compared to MCU and functions as a dominant-negative controller of the MCU complex. It is encoded by the ccdc109b gene and shares 50% similarity with MCU [44]. MCUb expression levels vary across tissues, suggesting its function as a tissue-specific modulator of MCU activity. The Essential MCU Regulator (EMRE) serves as a connecting element between the pore-forming MCU and the regulatory MICU1-MICU2 heterodimer [45]. It is crucial for MCU channel function and regulation, as it stabilizes the interaction between MCU and MICUs and mediates calcium-dependent activation of the channel [45]. EMRE not only stabilizes MCU and MICUs, but also actively plays a regulatory role in channel function [46]. Different models suggest that EMRE keeps MCU in an open conformation, ensuring calcium entry [47], or acts as a stabilizer of MCU-MICU1 interaction [48]. Physiological EMRE turnover is essential for correct assembly of the gatekeeping subunits and is regulated by mitochondrial AAA (m-AAA) protease [49,50]. MICU1 is a regulatory subunit of the mitochondrial calcium uniporter (mtCU), serving as a gatekeeper and binding to MCU through the DIME sequence [51]. At low levels of calcium, MICU1 restrains the movement of calcium, but at high levels, the calcium blocks the MCU-binding site in MICU1, leading to an increase in pore conductivity [52]. MICU1 forms homodimers or heterodimers with MICU2, and both have high affinities for calcium [52]. MICU1 has two structural domains containing EF-hand motifs, crucial for ion selectivity [50]. Its ability to differentiate between calcium and manganese is essential for channel regulation [53,54]. In vivo deletion of MICU1 causes lethality in Drosophila [55] and severe neurological and myopathic defects in mice, similar to MICU1-deficient patients [51]. The degradation of MICU1 is managed by Parkin and takes place in the proteasome. This process controls the composition and activity of MCU [56]. MICU1.1 is a variation of MICU1 that is produced through splicing and is expressed in skeletal muscle and, to a lesser degree, in the brain [57]. This form of MICU1 functions as a substitute regulator of the calcium uniporter in specific tissues and is believed to contribute to quick muscle contractions [57]. MICU1 has two isoforms, MICU2 and MICU3, which are believed to be located in the intermembrane space, but their specific roles remain unclear [58]. MICU2 and MICU1 form a heterodimer that utilizes the DIME motif, facing the intermembrane space [59]. According to electrophysiological recordings, MICU2 acts as a gatekeeper at low calcium levels, but has no effect when calcium levels are high [60]. The stability of MICU2 relies on the existence of MICU1 at the protein level. In cells where MICU1 expression has been silenced, there is a substantial decrease in the amount of MICU2 protein but not in MICU2 mRNA levels, suggesting a post-translational mechanism [61,62]. The loss of MCU gatekeeping in the absence of MICU1 might be due to the concurrent loss of MICU2 [61,62]. It appears that MICU3 has a limited role in the regulation process of the MCU complex, as it is mainly expressed in the central nervous system (CNS) [58]. However, the exact functions of MICU3 in the regulation of the MCU complex are still yet to be investigated. In MAMs, calcium transport from the ER to the mitochondria takes place through a complex made up of IP_3_R, Grp75, VDAC, and MCU, an overview of which is depicted in Figure 1.

MAMs, the junctions where mitochondria and ER connect, have a profound influence on cancer cell behavior, particularly in their capability to evade apoptosis or programmed cell death of cancer cells [63]. Changes in calcium signaling, a critical process managed by MAMs, are often associated with this evasion. MAMs harbor specific proteins like VDAC, IP_3_R, and GRP75, which have been identified as contributors to cancer cell survival and anti-apoptotic tendencies [64]. These proteins oversee calcium flux between the ER and mitochondria and can exhibit changed expression or activity in cancer cells, resulting in enhanced survivability and anti-apoptotic tendencies. As MAMs appear to be pivotal in cancer cell survival and anti-apoptotic behavior, researchers are exploring them as possible points of intervention for cancer treatment. By altering the makeup or activity of MAMs, it might be feasible to trigger apoptosis in cancer cells, thus curtailing their expansion and multiplication. Nonetheless, a deeper comprehension of MAMs’ intricate roles in cancer is required, as well as the development of efficient methods for their targeted manipulation in cancer treatment.

### 2.2. Calcium Activates Tricarboxylic Acid (TCA) Cycle

Mitochondria, which are central to managing apoptosis, depend on calcium for their function, including energy production and metabolic activities. Calcium in mitochondria influences energy metabolism by regulating pyruvate dehydrogenase (PDH) and TCA cycle enzymes including isocitrate dehydrogenase (IDH), and oxoglutarate dehydrogenase (OGDH) [65].

PDH is a component of the Pyruvate Dehydrogenase Complex (PDC), which transforms pyruvate into acetyl-CoA [66]. The operation of the PDC is controlled by factors such as end-products, phosphorylation and dephosphorylation through Pyruvate Dehydrogenase Kinases (PDKs) and Pyruvate Dehydrogenase Phosphatases (PDPs) [65]. Calcium triggers PDP1, which leads to the dephosphorylation of PDH and the conversion of pyruvate into acetyl-CoA [67,68].

IDH, a TCA cycle enzyme, transforms isocitrate into α-ketoglutarate [69]. The sensitivity of IDH to calcium binding is regulated by the ATP/ADP ratio, with a drop in the ratio leading to an increase in calcium binding and a decrease in the Km for isocitrate [70].

OGDH catalyzes the TCA cycle reaction that transforms α-ketoglutarate into succinyl-CoA [65]. Similar to IDH, calcium binding to OGDH reduces the Km for α-ketoglutarate [65]. 

In normal cells, when calcium in mitochondria surpasses normal levels, it can cause mitochondrial malfunction and stress, leading to the onset of apoptosis [71]. This happens due to the release of specific proteins like cytochrome c, which start a chain reaction resulting in cell death. Cancer cells often adapt their physiological processes to evade apoptosis. One method is by regulating mitochondrial calcium levels. They achieve this via overexpression of proteins that extrude calcium from mitochondria, suppressing proteins that permit calcium entrance, or modifying MAMs [72]. By inhibiting an overload of mitochondrial calcium, cancer cells can bypass the apoptosis trigger, thereby promoting their continued existence and proliferation. By preventing excessive mitochondrial calcium and thereby avoiding mitochondrial stress and dysfunction, cancer cells can maintain the normal operation of their mitochondria. This includes the tricarboxylic acid (TCA) cycle, a critical metabolic pathway that takes place within the mitochondria.

### 2.3. Calcium induces Reactive Oxygen Species (ROS) Production

Calcium primarily boosts ATP production by activating enzymes in the TCA cycle and oxidative phosphorylation within mitochondria. This surge in metabolic activity can result in a higher rate of respiratory chain electron leakage and an increase in ROS levels [73]. The metabolic conditions of mitochondria affect how calcium influences the levels of mitochondrial ROS. When the membrane potential is in a depolarized state (ATP synthesis), calcium’s impact on ROS generation is dependent on the calcium load and may stimulate it, and overloading mitochondria with calcium may increase ROS production [74,75]. Under normal conditions, up to 1% of electrons flowing to molecular oxygen through the electron transport chain might form superoxide [76]. Superoxide can be produced at various sites within mitochondria, including complex I (site IQ), complex III (site IIIQo), glycerol 3-phosphate dehydrogenase, the flavin in complex I (site IF), electron transferring flavoprotein:Q oxidoreductase (ETFQOR) of fatty acid beta-oxidation, and pyruvate and 2-oxoglutarate dehydrogenases. Notably, only site IIIQo (on complex III) and glycerol 3-phosphate dehydrogenase can release superoxide into the intermembrane space, indicating their importance in mitochondrial ROS release into the cytosol [77]. Calcium-mediated activation of the TCA cycle and ROS production is illustrated in Figure 2.

Calcium has a crucial role in normal cells, aiding in various mitochondrial functions, such as ATP production. However, excessive calcium levels can result in an overproduction of ROS, leading to cellular damage and apoptosis [78]. In cancer cells, the regulation of calcium and ROS is disrupted, with mechanisms in place to prevent excessive calcium accumulation in mitochondria, thus avoiding apoptosis [79]. This alteration can affect ROS levels, as reduced mitochondrial calcium can decrease ROS production. Despite elevated basal levels of ROS in cancer cells, they activate antioxidant pathways and develop survival strategies to counteract the effects of high ROS. Manipulating these processes for therapeutic purposes is complex, and ongoing research aims to better understand the relationship between ROS production, calcium signaling, and cancer.

### 2.4. Calcium and ROS Open Permeability Transition Pore (mPTP) 

The mPTP is a large conductance channel in the inner mitochondrial membrane, which is considered to have a major impact on cell death [80]. The opening of mPTP causes the collapse of the mitochondrial membrane potential and equalizes the concentration of water and small solutes smaller than 1.5 kD [81,82]. Although the exact composition of mPTP is not yet fully understood, some proteins like the adenine nucleotide translocator (ANT) and the F_1_F_0_ ATP synthase are thought to play a role in its formation and regulation. 

In the classical Adenine Nucleotide Translocator (ANT)/VDAC/cyclophilin D (CypD) model, mPTP formation was initially proposed to occur at connection sites between the inner and outer mitochondrial membranes [83]. The mPTP is highly conductive and non-selective, and its opening is triggered by calcium [84]. When CypD binds with ANT, it can trigger a transient, low-level opening of the mPTP, which promotes the expulsion of ROS from the mitochondria and helps maintain mitochondrial calcium homeostasis [84]. However, elevated levels of CypD or heightened activity can lead to excessive opening of the mPTP, causing an influx of substances with molecular weights below 1.5 kDa into the mitochondrial matrix [84]. This process leads to mitochondrial membrane depolarization, uncoupling of oxidative phosphorylation (OXPHOS), ATP depletion, and the release of proapoptotic factors, ultimately enhancing apoptosis [84]. Recent studies have shown that Bax can regulate the opening of the mPTP in collaboration with ANT [85]. The co-immunoprecipitation of Bax with ANT was observed, and its overexpression caused cell death only when ANT was present. Both ANT and Bax were crucial for the formation of the mPTP channel in artificial membranes as a result of atractyloside [85]. 

The F_1_F_0_-ATPase dimer model proposes that mPTP forms at the junction of two F_1_F_0_-ATPase proteins [86]. Studies conducted recently have indicated that the interaction between CypD and the oligomycin-sensitive conferring protein (OSCP) of F_1_F_0_-ATPase results in the formation of mPTP [87]. It is widely acknowledged that the binding of Cyclosporin A (CsA) to the ATP synthase OSCP subunit can decrease mPTP formation by inhibiting the binding of CypD. Recent studies have also shown that calcium binds to Thr 163 of F_1_F_0_-ATPase to open mPTP and found that Thr 163 mutations exhibited a decrease in calcium sensitivity to mPTP opening [88].

The F_1_F_0_-ATPase C-Ring model suggests that the c-subunit of the F_1_F_0_-ATP synthase plays important role in mPTP formation. Alavian et al. discovered that Bcl-xL binds to the ATP synthase at the β-subunit, modulating its activity [89]. They also found that purified c-subunits produced a voltage-sensitive channel when reconstituted into liposomes [90]. High calcium levels caused the c-ring to detach from the F_1_ subunit [91]. According to this theory, under conditions favoring mPTP formation, the F_1_ portion of the F_1_F_0_-ATP synthase separates from the F_0_ portion, enabling the channel pore in the c-subunit to allow solute passage. Under physiological conditions, the β-subunit of the F_1_ subunit helps prevent the flow of substances across the pore [90]. 

ROS do not directly affect mPTP formation, but they can contribute to the opening of the mPTP and subsequent mitochondrial dysfunction. ROS promotes an increase in cytosolic calcium levels. High levels of calcium can be taken up by the mitochondria, leading to calcium overload. This can trigger the opening of the mPTP. 

In normal cells, high calcium levels can trigger the opening of the mPTP, leading to mitochondrial membrane permeabilization, release of apoptotic factors, and potential cell death [91]. However, cancer cells have altered regulation of the mPTP, developing mechanisms to prevent excessive calcium accumulation in mitochondria, thus reducing mPTP opening and inhibiting apoptosis [92]. This dysregulation helps cancer cells survive and resist cell death, posing a significant obstacle in cancer treatment. Research is being performed with the aim of understanding how this dysregulation impacts cancer cell survival and resistance to apoptosis, providing insights for developing strategies to overcome this challenge in cancer therapy. The mechanism of mPTP formation by calcium and ROS is shown in Figure 3.

### 2.5. mPTP Induces Cytochrome c Release

Under standard biological conditions, the inner mitochondrial membrane is largely impermeable, allowing only a limited number of specific metabolites and ions to traverse the membranes. In well-functioning cells, the mPTP is usually closed and opens exclusively when triggered by external factors. The opening of mPTP can be triggered by calcium overload in the mitochondrial matrix and excessive reactive oxygen species (ROS) production, which leads to oxidative stress [93]. When mPTP opens, mitochondrial permeability rises, permitting the unregulated passage of solutes, such as water, molecules, and ions, into the mitochondrial matrix. As this progresses, the opening of mPTP leads to the enlargement of mitochondria, the disruption of the outer mitochondrial membrane (OMM), and compromised electron transport chain (ETC) function. Consequently, this causes a substantial release of reactive oxygen species (ROS), cytochrome c, and SMAC/DIABLO [93]. Additionally, the increased mitochondrial permeability leads to a loss of mitochondrial membrane potential (ΔΨM) and a subsequent decrease in cellular mitochondrial ATP. It has been suggested that ROS can facilitate the formation of disulfide bridges between BAX monomers, leading to conformational changes that enable dimerization or channel formation by translocated BAX in the outer mitochondrial membrane [94]. In the cytosol, cytochrome c participates in the formation of a large protein complex known as the apoptosome, which activates caspase-9, a key enzyme in the apoptotic pathway [95,96]. The activation of caspase-9 sets off a series of events, which involve the stimulation of downstream effector caspases, culminating in a regulated process of cell disassembly. Consequently, the opening of mPTP instigates the activation of intrinsic apoptotic mechanisms, enhancing the cancer cell death induced by chemotherapy.

SMAC, also known as DIABLO, is a pro-apoptotic protein with a critical role in managing apoptosis or programmed cell death. Found in the mitochondria, SMAC/DIABLO gets released into the cytosol during apoptosis as a response to various cellular stress factors such as DNA damage, oxidative stress, or endoplasmic reticulum stress, which lead to the permeabilization of the mitochondrial outer membrane. The primary function of SMAC/DIABLO is to facilitate apoptosis by counteracting inhibitor of apoptosis proteins (IAPs) [97,98]. IAPs are a group of proteins that prevent apoptosis by binding to and suppressing the activity of caspases, a collection of protease enzymes vital for executing apoptosis. By interacting with IAPs, SMAC/DIABLO stops them from restraining caspases, thereby enabling caspase activation and the initiation of the apoptotic process. The release of cytochrome c via mPTP is shown in Figure 4.

## 3. The Role of Calcium in ER Stress-Mediated Apoptosis

Calcium has a significant function in apoptosis mediated by endoplasmic reticulum (ER) stress. An imbalance in calcium levels within the ER can interfere with the protein folding process, resulting in the buildup of improperly folded or unfolded proteins, which subsequently induces ER stress.

### 3.1. Unfolded Protein Response (UPR) and ER Stress

The ER serves as the primary organelle for signal transduction, detecting homeostatic changes, and offering feedback to other cellular components [99]. Proteins usually fold into their tertiary and quaternary structures within the ER [100]. Disruptions in cellular ATP levels, calcium concentrations, or redox state can hinder the ER’s protein-folding capabilities, leading to the accumulation of unfolded proteins and the onset of ER stress [101]. Heightened protein traffic and the accumulation of unfolded protein aggregates can also induce ER stress [102]. To counter this, the UPR is activated, an adaptive mechanism aimed at reestablishing protein homeostasis [103]. The UPR is regulated by three ER-localized proteins: inositol requiring protein-1 (IRE1), protein kinase RNA-like ER kinase (PERK), and activating transcription factor-6 (ATF6). These proteins bind to the ER chaperone binding-immunoglobulin protein (BiP) and remain inactive under unstressed conditions [104]. When ER stress occurs, they separate from BiP and become active.

Under normal conditions, IRE1 binds to the chaperone protein BiP (Grp78). In the presence of misfolded or unfolded proteins within the ER, BiP tends to associate with these proteins to assist in achieving correct folding or to promote their degradation [105]. As a result, BiP dissociates from IRE1, allowing the luminal domains of nearby IRE1 molecules to interact with each other. This interaction promotes IRE1 oligomerization and trans-autophosphorylation at serine 724 residue, leading to its activation. When activated, IRE1 splices X-box-binding protein 1 (XBP1) mRNA, generating spliced XBP1 (sXBP1) [105,106]. sXBP1 upregulates UPR-related genes including protein folding, protein translocation to the ER, and ER-associated degradation (ERAD) [107,108]. IRE1 also recruits tumor necrosis factor receptor (TNFR)-associated factor-2 (TRAF2) and activates apoptosis-signaling kinase 1 (ASK1) [109]. ASK1 activation leads to c-Jun N-terminal protein kinase (JNK) and p38 mitogen-activated protein kinase (MAPK) activation [110]. Activated JNK translocates to the mitochondrial membrane, activating Bcl-2 interacting protein (Bim) and inhibiting Bcl-2, while p38 MAPK phosphorylation activates transcriptional factor C/EBP homologous protein (CHOP), increasing Bim and death receptor 5 (DR5) expression and simultaneously decreasing Bcl-2 expression, initiating apoptosis [111]. Bcl-2-associated X protein (Bax) and Bcl-2 homologous antagonist killer protein (Bak) have the ability to interact with and activate IRE1, as well as stimulate IP_3_Rs, leading to the release of calcium from the ER [106]. 

PERK, a type I transmembrane protein located in the ER, detects ER stress and reduces mRNA translation [112]. When PERK is activated, it prevents the entry of newly synthesized proteins into the stressed ER compartment by activating eukaryotic initiation factor 2α (eIF2α) via serine 51 phosphorylation [113]. This process inhibits eIF2B, a guanine nucleotide exchange factor complex responsible for recycling eIF2α to its active GTP-bound state [114], thus decreasing misfolded protein overload and relieving ER stress. Phosphorylation of eIF2α also facilitates the translation of genes dependent on the UPR, such as ATF4, which possess multiple upstream open reading frames [115]. ATF4 stimulates the expression of ER stress target genes, including CHOP, growth arrest and DNA-damage-inducible 34 (GADD34), and ATF3 [116].

ATF6, a type II transmembrane protein found in the ER, detaches from BiP and relocates to the Golgi compartment under ER stress conditions for additional proteolytic processing. Two Golgi-based enzymes, site-1 protease (S1P) and site-2 protease (S2P), participate in the proteolytic cleavage of the full-length 90-kDa ATF6 [117]. Subsequently, the cleaved N-terminal cytosolic domain of the 50 kDa cytosolic basic leucine zipper (bZIP) migrates to the nucleus and binds to ATF/cAMP response elements (CRE) and ER stress-response elements (ERSE-1) to activate the transcription of target proteins, such as BiP, XBP-1, and CHOP [118].

During prolonged ER stress, IRE1, PERK, and ATF6 can induce pro-apoptotic signaling through the activation of CHOP, which subsequently leads to the initiation of apoptosis [119]. 

### 3.2. Calcium Activates ER Stress

An imbalance in calcium concentration within the ER can interfere with protein folding, resulting in the accumulation of misfolded or unfolded proteins, which in turn triggers ER stress [120]. This imbalance may arise from various factors, including changes in calcium channel activity, exposure to toxins or stressors, or alterations in the function of calcium-binding chaperone proteins. The correct folding of proteins in the ER relies on calcium-dependent chaperone proteins like calnexin and calreticulin, which bind to newly formed proteins and aid in their proper folding [121]. However, when calcium levels are disrupted, these chaperone proteins may not function effectively, causing misfolded proteins to accumulate and initiating the UPR, which seeks to reestablish ER homeostasis. Apoptosis is triggered by ER stress when the UPR is unable to reestablish ER homeostasis, and the stress becomes persistent or intense [122]. Normal cells tightly regulate calcium homeostasis, and ER stress induced by calcium occurs in specific circumstances. Conversely, cancer cells frequently display disrupted calcium signaling and modified responses to ER stress. This allows cancer cells to adapt to ER stress and facilitate cell survival, contributing to the advancement of tumors and resistance to therapy [104]. Calcium-induced ER stress signaling is shown in Figure 5.

## 4. The Role of Calcium in Cytosolic Protein-Mediated Apoptosis

Cytosolic calcium levels are tightly regulated, and an imbalance in calcium concentrations can trigger apoptosis through various mechanisms involving cytosolic proteins. Here are some ways calcium contributes to cytosolic protein-mediated apoptosis.

### 4.1. Calcium/CAMK II/JNK/Fas Pathway

The calcium-regulated CaMKII/JNK/Fas pathway is a critical signaling cascade involved in controlling apoptosis. This process begins with elevated intracellular calcium levels, leading to the activation of CaMKII, a serine/threonine protein kinase. CaMKII is activated when it binds to calmodulin, a calcium-binding protein, in the presence of increased calcium concentrations [123]. Once activated, CaMKII phosphorylates a variety of substrates, such as transcription factors, ion channels, and other kinases, thereby influencing their functions. A key downstream target of CaMKII is the JNK signaling pathway. CaMKII activates the JNK pathway by stimulating upstream kinases like MKK4/7 [123]. Following activation, MKK4/7 phosphorylates and activates JNK, a stress-activated protein kinase involved in several cellular processes, including apoptosis. Activated JNK phosphorylates and activates multiple transcription factors, among them c-Jun, a component of the AP-1 complex. The AP-1 complex is essential for controlling the expression of various genes, including those associated with apoptosis. Fas, an apoptosis-related gene regulated by JNK, is a cell surface receptor that is part of the TNF receptor superfamily. When activated, Fas forms a death-inducing signaling complex (DISC) with FADD and procaspase-8, leading to caspase-8 activation. The subsequent caspase cascade ultimately results in apoptosis. In normal cells, calcium-induced activation of CAMK II can trigger the activation of JNK, leading to apoptosis through different downstream targets, including the Fas receptor. 

However, in cancer cells, there can be disruptions in calcium signaling, resulting in abnormal CAMK II activation [124]. Additionally, alterations in the expression and functionality of the Fas in cancer cells can impact its involvement in apoptotic signaling [125]. These differences contribute to distinct apoptotic responses between normal cells and cancer cells.

### 4.2. Calcium/Calcineurin/Bcl-2 Pathway

Calcineurin, sometimes referred to as protein phosphatase 2B (PP2B), is a serine/threonine protein phosphatase that relies on calcium and calmodulin for activation [126,127,128]. It plays a critical role in various cellular processes, including cell signaling, immune response, and the regulation of apoptosis. The activation of calcineurin occurs when there is an increase in intracellular calcium levels, which causes calcium ions to bind to calmodulin. This calcium/calmodulin complex then interacts with calcineurin, triggering a conformational change that activates its phosphatase activity [127,128]. Once activated, calcineurin dephosphorylates a range of substrates, including transcription factors and other proteins, modulating their functions. One of the targets of activated by calcineurin is the protein BAD. Dephosphorylation of BAD by calcineurin causes it to detach from the 14-3-3 protein and relocate to the outer mitochondrial membrane (OMM) [129]. BAD then forms dimers with anti-apoptotic proteins Bcl-2 and Bcl-xL, suppressing pro-survival signals. This process prompts BAX to translocate to the OMM, initiating the formation of mPTP. The formation of mPTP accelerates the apoptotic cascade by releasing cytochrome c from the mitochondria, ultimately leading to cell death execution by caspases 3, 6, and 7.

### 4.3. Calpain/Caspases Pathway

Calpain, a family of calcium-dependent cysteine proteases, has important roles in various cellular processes, such as signal transduction, cytoskeletal remodeling, cell differentiation, and apoptosis [130]. These proteases are activated when intracellular calcium levels increase, enabling them to cleave specific target proteins through their proteolytic functions. Caspase-12 is an ER-associated caspase. During ER stress, calpain becomes activated by elevated intracellular calcium levels and cleaves the inactive pro-caspase-12, releasing active caspase-12 from the ER membrane [131]. Active caspase-12 participates in the apoptotic signaling cascade by cleaving and activating caspase-9, which subsequently activates executioner caspases such as caspase-3, caspase-6, and caspase-7. These caspases then cleave various cellular substrates, leading to the characteristic features of apoptosis.

Calpains also cleave pro-apoptotic proteins like Bid. The cleaved Bid (tBid) translocates to the mitochondria, where it interacts with Bax and Bak, resulting in mitochondrial outer membrane permeabilization (MOMP) [132]. MOMP allows cytochrome c and other pro-apoptotic factors to be released into the cytosol. Furthermore, calpains can cleave and inactivate anti-apoptotic proteins such as Bcl-2, shifting the balance in favor of apoptosis [133]. 

### 4.4. PKC/PDK/ERK Pathway

Protein kinase C (PKC) consists of a group of serine/threonine kinases activated by various signals, including growth factors, hormones, and neurotransmitters [134]. PKC is typically activated by binding to diacylglycerol (DAG), a lipid-derived secondary messenger, and elevated intracellular calcium levels. A downstream target of PKC is NADPH oxidase, an enzyme complex composed of multiple subunits that produce reactive oxygen species (ROS), such as superoxide anion (O_2_^−^) and hydrogen peroxide (H_2_O_2_). PKC activates NADPH oxidase by phosphorylating its regulatory subunits, causing the enzyme complex to assemble and become active [135]. When NADPH oxidase is activated, it generates ROS, which are capable of damaging cellular components, including proteins, lipids, and DNA. Normally, ROS production is carefully regulated, and ROS serve as signaling molecules that control various cellular processes like cell proliferation, differentiation, and survival. In the context of apoptosis, ROS can induce cell death through multiple mechanisms. For example, ROS can directly damage mitochondrial components, leading to mitochondrial outer membrane permeabilization and the subsequent release of pro-apoptotic factors like cytochrome c, which triggers the intrinsic apoptotic pathway. Additionally, ROS can activate stress kinases such as JNK and p38 MAPK, which can phosphorylate and modulate the activity of pro- and anti-apoptotic proteins, tipping the balance towards apoptosis. The signaling pathways involving calcium-regulated cytoplasmic proteins in apoptosis are depicted in Figure 6.

## 5. Conclusions

The regulation of calcium signaling is critical for the induction and execution of apoptosis. Imbalances in calcium signaling have been connected to various health issues, making it a significant target for creating new therapeutic approaches. The intricate regulation of calcium signaling in apoptosis involves the coordinated activity of several ion channels, transporters, and downstream effectors. Understanding the precise mechanisms by which calcium signaling regulates apoptosis is crucial for developing targeted therapies to treat various diseases. In conclusion, a comprehensive summary of the various aspects discussed in this review can be found in Figure 7. This figure highlights the interplay between calcium and its role in apoptosis through different signaling pathways, cellular compartments, and protein interactions, providing a visual representation of the complex nature of calcium-mediated apoptotic processes.

This review paper focused on cancer-specific calcium-induced apoptosis. Both general and cancer-specific apoptosis processes involve controlling intracellular calcium levels to induce cell death and play critical roles in maintaining cellular homeostasis and removing damaged or unwanted cells. However, these processes exhibit distinct mechanisms and contexts.

General calcium-induced apoptosis is a normal physiological response to increased intracellular calcium levels, essential for maintaining tissue homeostasis by eliminating damaged or unneeded cells. The elevated intracellular calcium triggers various signaling pathways, such as the mitochondria-mediated intrinsic pathway and the death receptor-mediated extrinsic pathway, leading to caspase activation and cell dismantling.

In contrast, cancer-specific calcium-induced apoptosis seeks to selectively induce cell death in cancer cells without affecting healthy cells. This selectivity can be achieved by targeting specific calcium channels or signaling pathways deregulated in cancer cells. For example, some cancer cells may overexpress certain calcium channels (TRPC, TRPV, TRPM), making them more susceptible to calcium-induced apoptosis. Additionally, cancer cells may have alterations in calcium signaling pathways that can be targeted to induce apoptosis. Such alterations may include changes in the expression of calcium-binding proteins, dysregulated calcium release from the endoplasmic reticulum (ER), or aberrant activation of calcium-dependent kinases and phosphatases.

The primary advantage of cancer-specific calcium-induced apoptosis is its potential as a therapeutic strategy for cancer treatment. By identifying the unique calcium channels and signaling pathways in cancer cells, researchers can develop drugs that selectively induce apoptosis in cancer cells, minimizing damage to healthy cells. This approach may lead to more effective and less toxic cancer therapies.

In summary, while general calcium-induced apoptosis is a physiological process that occurs in various cell types, cancer-specific calcium-induced apoptosis aims to selectively induce cell death in cancer cells by exploiting differences in calcium regulation or signaling pathways. Understanding these differences is crucial for developing targeted therapies that selectively eliminate cancer cells, improving overall cancer treatment outcomes.

## Figures and Tables

**Figure 1 ijms-24-08982-f001:**
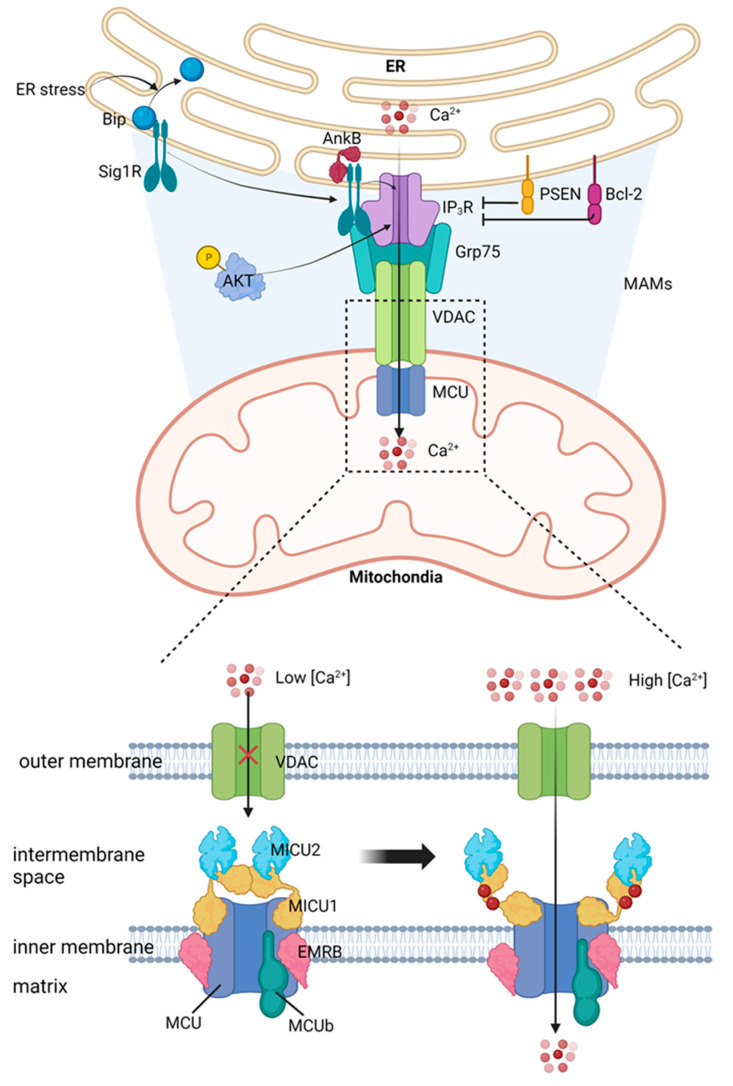
Calcium transport to mitochondria from ER is regulated by several proteins, including IP_3_R and VDAC. Both of these proteins play important roles in controlling calcium movement between different cellular compartments.

**Figure 2 ijms-24-08982-f002:**
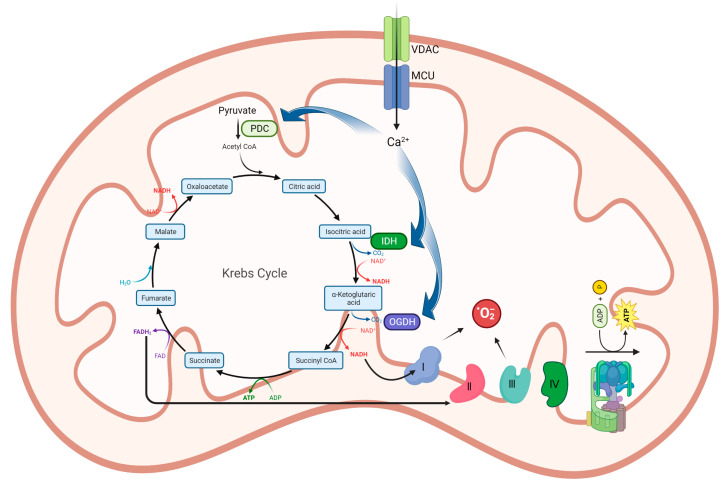
Calcium serves as a coenzyme for enzymes participating in the TCA cycle. By increasing the activity of the TCA cycle, calcium accelerates the generation of NADH and FADH_2_, ultimately boosting the electron transport chain and stimulating ROS production.

**Figure 3 ijms-24-08982-f003:**
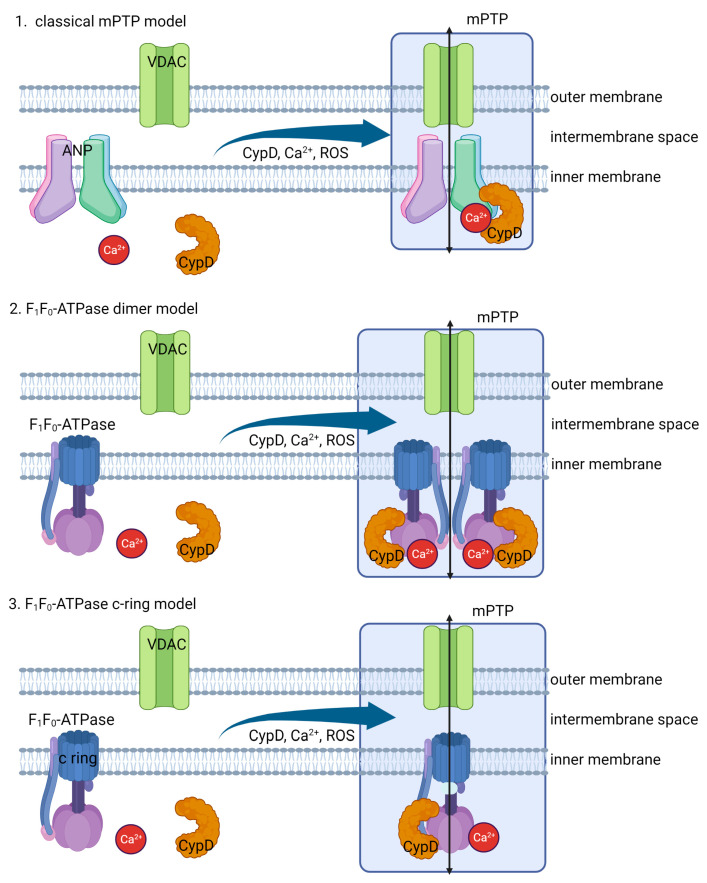
Three models of mPTP formation. ANT/VDAC/CypD model suggests that mPTP formation involves adenine nucleotide translocase (ANT), voltage-dependent anion channel (VDAC), and cyclophilin D (CypD). In this scenario, ANT, VDAC, and CypD interact to create a channel that allows for the passage of calcium and solutes across the mitochondrial membrane, leading to permeability transition. F_1_F_0_-ATPase dimer model proposes that mPTP is formed by the dimerization of F_1_F_0_-ATPase complexes. In this case, the dimerization of these complexes creates a channel that allows for the movement of calcium across the mitochondrial membrane, leading to permeability transition. In F_1_F_0_-ATPase C-Ring model, the mPTP is formed by the F_1_F_0_-ATPase C-ring subunit. The C-ring acts as a channel that permits the movement of calcium across the mitochondrial membrane, leading to permeability transition.

**Figure 4 ijms-24-08982-f004:**
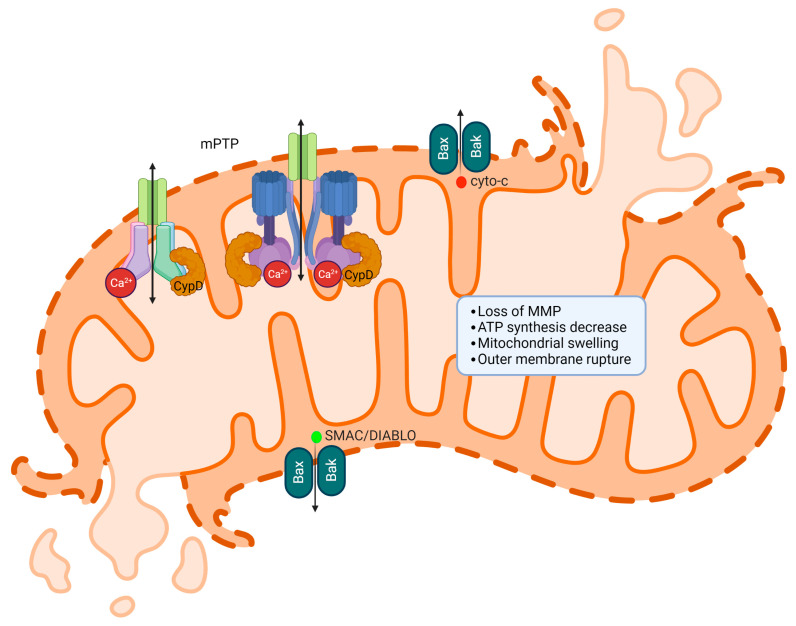
The release of cytochrome c and SMAC/DIABLO occurs via mPTP. When mPTP opens, it disrupts the mitochondrial membrane potential, allowing ions and solutes to cross the membrane. This results in mitochondrial swelling and outer mitochondrial membrane rupture, which subsequently releases cytochrome c and SMAC/DIABLO into the cytosol. Once in the cytosol, cytochrome c initiates the apoptotic cascade by activating caspases, leading to programmed cell death and SMAC/DIABLO inhibits IAPs.

**Figure 5 ijms-24-08982-f005:**
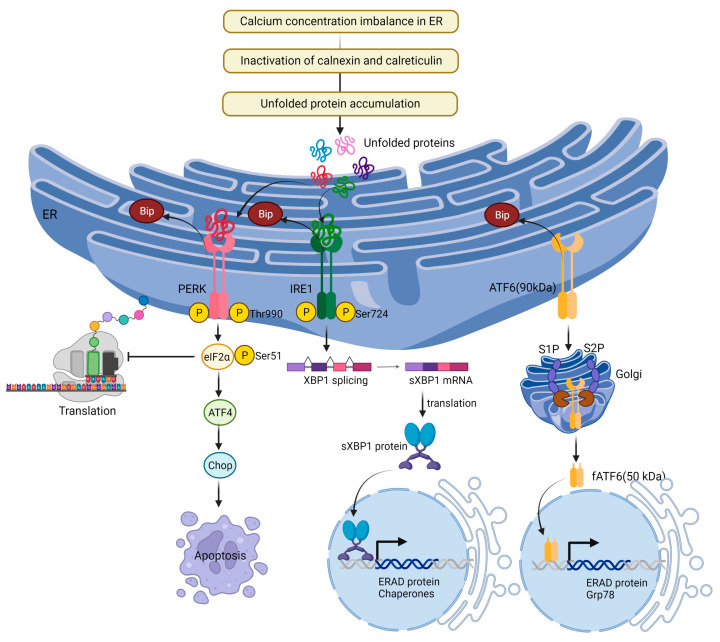
Calcium-induced ER stress signaling by UPR. ER stress occurs when misfolded or unfolded proteins accumulate in the ER, disrupting its normal function. The UPR is mediated by three primary ER transmembrane sensors: IRE1, PERK, and ATF6. When activated, these sensors trigger downstream signaling events to help the cell adapt to ER stress. If the stress is too severe or prolonged, the UPR shifts from a pro-survival to a pro-apoptotic response, ultimately leading to cell death.

**Figure 6 ijms-24-08982-f006:**
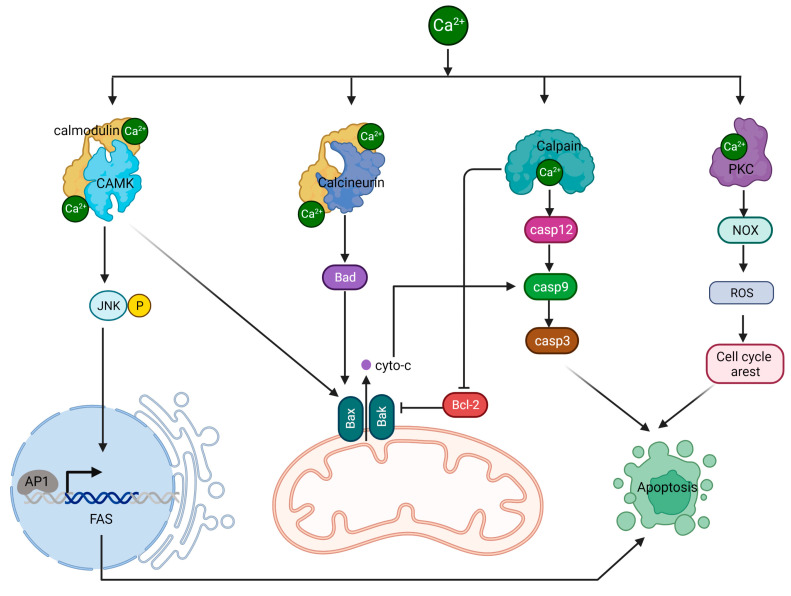
The regulation of apoptosis by calcium. Elevated calcium levels contribute to apoptosis activation by modulating the activities of several proteins, including CAMK, calcineurin, calpain, and PKC. These proteins participate in various signaling pathways and cellular processes, ultimately leading to cell death when calcium concentrations become excessive.

**Figure 7 ijms-24-08982-f007:**
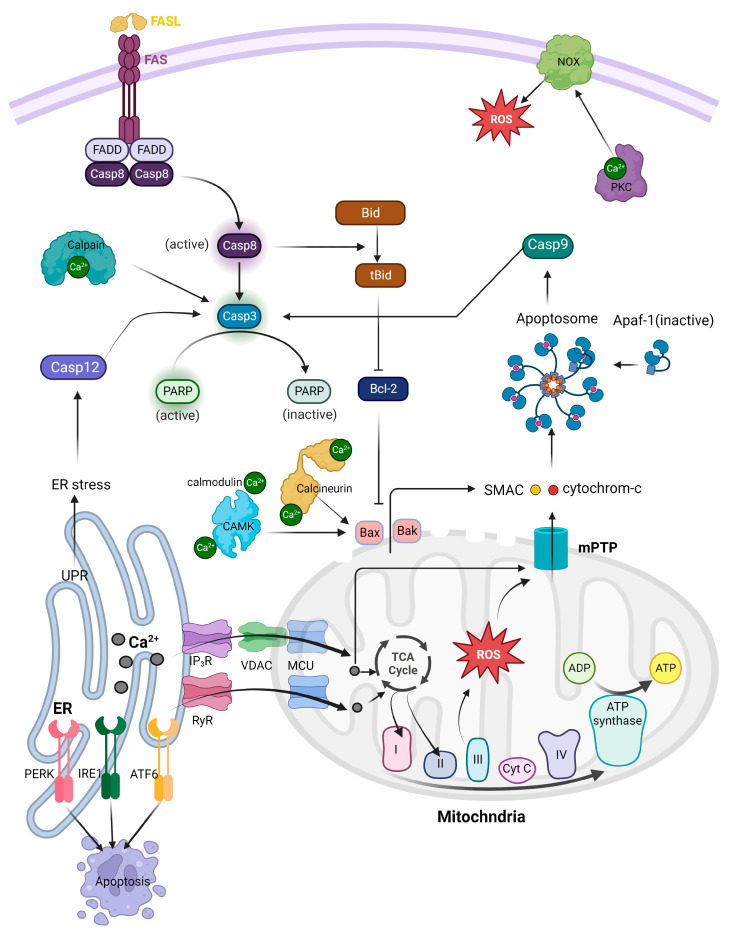
Comprehensive summary of the role of calcium in apoptosis.

## Data Availability

Not applicable.

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
