# Peer review of "Calcium’s Role in Orchestrating Cancer Apoptosis: Mitochondrial-Centric Perspective"

_ijms, 2023, doi:10.3390/ijms24108982_

Round 1

Reviewer 1 Report

The author has explained about the calcium signaling with a focus on mitochondria in orchestrating cancer. Aim of this manuscript is interesting, but there are critical concerns that the authors should address as follows.

Comment 1

Some parts of this review are too basic and general and should be simplified. For example, most of Fig. 1 and Fig. 2 are already common content. Furthermore, I believe that this would emphasize what the author wants to argue.

Comment 2

The author titles the manuscript "Calcium's Role in Orchestrating Cancer Cell Death," but it was not clear which part of the manuscript is cancer cell-specific cell death. It would be better to describe the difference between general calcium-induced apoptosis and cancer-specific calcium-induced apoptosis so that it is clear.

Author Response

Comment 1

Some parts of this review are too basic and general and should be simplified. For example, most of Fig. 1 and Fig. 2 are already common content. Furthermore, I believe that this would emphasize what the author wants to argue.

⟶ As you pointed out, Figure 1 and Figure 2 contain common content that is fundamental to the topic being discussed. As you're likely aware, open access (OA) papers can be accessed online by anyone wishing to read, download, or use them, without facing financial, legal, or technical barriers. My goal in this review paper is to ensure that it can be easily comprehended not just by experts in the field, but also by graduate students, college students, and the general public. I am of the opinion that incorporating such common content helps provide the necessary context and background for readers who may not be well-versed in the subject matter.

Nonetheless, based on your suggestion, I understand that excluding general background information and placing greater emphasis on the main content could enhance the overall quality and focus of the paper. If you strongly believe that this background information should be omitted, please don't hesitate to reach out to me again, and I will gladly make the necessary adjustments.

Comment 2

The author titles the manuscript "Calcium's Role in Orchestrating Cancer Cell Death," but it was not clear which part of the manuscript is cancer cell-specific cell death. It would be better to describe the difference between general calcium-induced apoptosis and cancer-specific calcium-induced apoptosis so that it is clear.

⟶ Thank you very much for your valuable feedback. Information regarding cancer-specific calcium-induced apoptosis has been incorporated into the discussion section.

Reviewer 2 Report

It’s well writing. I just think if it will better to describe ER stress mediated apoptosis then mitochondrial, since which including MAM, and Ca2+ from ER.

And some other compartments are also involve in Ca2+-mediated cell death, for example lysosome, the author can discuss a little of the relationship between them.  

REF

Chu H, Qin Z, Ma J, Xie Y, Shi H, Gu J, Shi B. Calcium-Sensing Receptor (CaSR)-Mediated Intracellular Communication in Cardiovascular Diseases. Cells. 2022 Sep 30;11(19):3075. doi: 10.3390/cells11193075. PMID: 36231037

Ning B, Guo C, Kong A, Li K, Xie Y, Shi H, Gu J. Calcium Signaling Mediates Cell Death and Crosstalk with Autophagy in Kidney Disease. Cells. 2021 Nov 17;10(11):3204. doi: 10.3390/cells10113204.PMID: 34831428

Author Response

It’s well writing. I just think if it will better to describe ER stress mediated apoptosis then mitochondrial, since which including MAM, and Ca2+ from ER.

⟶ Thank you for your suggestion. In this review paper, we emphasize the Mitochondria-associated membranes (MAMs) derived calcium-induced apoptosis for several reasons.

Firstly, MAMs-derived calcium-induced apoptosis represents a relatively novel and emerging research area compared to the well-explored role of calcium in ER stress-mediated apoptosis. By focusing on MAMs, we can offer fresh insights and stimulate further investigation into their role in apoptosis regulation, potentially leading to the identification of new therapeutic targets.

Secondly, MAMs play a critical role in maintaining communication between the mitochondria and the endoplasmic reticulum (ER). A review paper that highlights MAMs-derived calcium-induced apoptosis can enhance our understanding of the intricate interplay between these organelles in cellular signaling, metabolism, and apoptosis regulation.

Thirdly, MAMs dysfunction has been linked to numerous pathological conditions, including neurodegenerative diseases, metabolic disorders, and cancer. A review paper concentrating on MAMs-derived calcium-induced apoptosis can help consolidate existing knowledge on MAMs' role in these diseases, potentially guiding the development of new diagnostic and therapeutic strategies.

Fourthly, the field of MAMs-derived calcium-induced apoptosis is still emerging, and there may be many undiscovered molecular targets within this pathway. A review paper that focuses on MAMs can motivate researchers to explore these potential targets, possibly leading to the development of innovative therapeutic interventions.

Lastly, while the role of calcium in ER stress-mediated apoptosis is well-established, our understanding of MAMs-derived calcium-induced apoptosis is still evolving. A review paper emphasizing MAMs can help bridge gaps in current knowledge and inspire further research into the complex interplay between calcium signaling, mitochondrial function, and apoptosis regulation.

If you believe that additional information on the role of calcium in ER stress-mediated apoptosis is necessary, please feel free to contact me again.

And some other compartments are also involve in Ca2+-mediated cell death, for example lysosome, the author can discuss a little of the relationship between them.  

REF

Chu H, Qin Z, Ma J, Xie Y, Shi H, Gu J, Shi B. Calcium-Sensing Receptor (CaSR)-Mediated Intracellular Communication in Cardiovascular Diseases. Cells. 2022 Sep 30;11(19):3075. doi: 10.3390/cells11193075. PMID: 36231037

Ning B, Guo C, Kong A, Li K, Xie Y, Shi H, Gu J. Calcium Signaling Mediates Cell Death and Crosstalk with Autophagy in Kidney Disease. Cells. 2021 Nov 17;10(11):3204. doi: 10.3390/cells10113204.PMID: 34831428

⟶ Thank you for your valuable comments and suggested references. In the initial review paper, I intended to discuss the role of calcium in the 13 types of Regulated Cell Death (RCD), including Apoptosis, Necroptosis, Pyroptosis, Ferroptosis, Autophagy-dependent cell death, Parthanatos, Lysosome-dependent cell death, Immunogenic cell death (ICD), NETotic cell death, Oxeiptosis, Alkaliptosis, Entotic cell death, and MPT-driven necrosis. However, the content became too extensive, so I decided to focus on the role of calcium in apoptosis. The role of calcium in the remaining 12 types of cell death will be addressed in another review. Thus, the relationship between calcium and other cellular compartments will be explored in a subsequent review paper currently in preparation, such as the role of calcium in lysosome-dependent cell death and the role of calcium in autophagy-dependent cell death. I felt that using the term "cell death" in the paper title could be confusing, so I changed it to "apoptosis" to avoid any misinterpretation.

Round 2

Reviewer 1 Report

I read your reply.

Regarding comment 1, I believe that the introduction should be simplified more, instead of deleting all the introductions. If existing knowledge is to be described in detail, it should be in book form and is not appropriate for a review article.

Then, regarding comment 2, I think it is good that you mentioned the difference between ordinary cells and cancer cells in the conclusion. However, as far as I read, there seemed to be very few places in the text where the difference between cancer cells and cancer cells was mentioned. If the title is "Calcium's Role in Orchestrating Cancer Apoptosis," it should clarify the characteristics of apoptosis by cancer cells, explain calcium signaling specific to cancer cells, or focus the story on cancer cells. The story should focus on cancer cells.

Author Response

I read your reply.

Regarding comment 1, I believe that the introduction should be simplified more, instead of deleting all the introductions. If existing knowledge is to be described in detail, it should be in book form and is not appropriate for a review article.

⟶ The contents of the introduction were simplified and the figures in figures 1 and 2 were omitted.

Then, regarding comment 2, I think it is good that you mentioned the difference between ordinary cells and cancer cells in the conclusion. However, as far as I read, there seemed to be very few places in the text where the difference between cancer cells and cancer cells was mentioned. If the title is "Calcium's Role in Orchestrating Cancer Apoptosis," it should clarify the characteristics of apoptosis by cancer cells, explain calcium signaling specific to cancer cells, or focus the story on cancer cells. The story should focus on cancer cells.

⟶ Added role of calcium in cancer cells.

Round 3

Reviewer 1 Report

The addition of the description of apoptosis and calcium signaling in cancer cells is sufficient to meet the purpose of this review. However, there is one point that needs to be improved. Please be sure to include the citations for the sections you have added this time, as they are not listed.

Author Response

Thank you very much for your sincere review. References have been added as per your suggestion.